# Quantifying probabilistic robustness of tree-based classifiers against natural distortions

## Abstract

The concept of trustworthy AI has gained widespread attention lately. One of the aspects relevant to trustworthy AI is robustness of ML models. In this study, we show how to probabilistically quantify robustness against naturally occurring distortions of input data for tree-based classifiers under the assumption that the natural distortions can be described by multivariate probability distributions that can be transformed to multivariate normal distributions. The idea is to extract the decision rules of a trained tree-based classifier, separate the feature space into non-overlapping regions and determine the probability that a data sample with distortion returns its predicted label. The approach is based on the recently introduced measure of "real-world-robustness", which works for all black box classifiers, but is only an approximation~~and only works if the input dimension is not too high~~, whereas our proposed method gives an exact measure.

## 1 Introduction

Robustness of machine learning models is a recently widely investigated topic ~~.~~ and a variety of robustness measures, e.g., closest counterfactual, have been developed. Scher & Trügler (2022) introduced the term *real-world-robustness*, which describes a general framework to compute the robustness of individual predictions of a trained machine learning model against natural distortions, e.g., data-processing errors, noise or measurement errors in the input data. The real-world-robustness $\mathbf{R}_\mu$ of the prediction $f$ of an $N$-dimensional data sample $\mu$ with distortion $p_\mu(\vec{x})$ given as a probability density function (PDF) is defined via a binary function $f'$,

$$f'(\mu, \epsilon) = \begin{cases} 0, & f(\mu + \epsilon) = f(\mu) \\ 1, & f(\mu + \epsilon) \neq f(\mu) \end{cases} \tag{1}$$

with $\epsilon \sim p_\mu(\vec{x})$ and an integral that determines the probability $P$ for a different prediction compared to the data sample $\mu$,

$$P(f(\mu + \epsilon) \neq f(\mu)) = \frac{\int\limits_{\mathbb{R}^N} f'(\mu, \epsilon) \, p_\mu(\vec{x}) \, d\epsilon}{\int\limits_{\mathbb{R}^N} f'(\mu, \epsilon) \, d\epsilon}. \tag{2}$$

The real-world-robustness $\mathbf{R}_\mu$ of the data sample $\mu$ with distortion $\epsilon \sim p_\mu(\vec{x})$ is then computed by

$$\mathbf{R}_\mu = 1 - P.$$

It is therefore a *probabilistic* measure. In words, real-world-robustness is the probability that the prediction (classification) of an input sample does not change under the given uncertainty of the input sample. Scher & Trügler (2022) showed how this probabilistic robustness measure can approximately be computed for any black-box classifier with a Monte-Carlo based method, under the constraint that the input feature space is not too high dimensional. They additionally provide a detailed discussion for the justification of

this definition of real-world-robustness, and how it differs from other robustness measures. The practical relevance of this robustness measure are settings in which the uncertainty of input samples is known (or at least approximately known), and in which one needs a measure of how likely it is that a classification will change due to the random error in the input data. This could for example be applications that use multivariate sensor data with measurement noise (e.g., temperature measurements at different locations) or where medical measuring devices (e.g., Glucose meter) are being used.

In this paper, we quantify probabilistic robustness by exactly computing the measure from Scher & Trügler (2022) for tree-based classifiers (Decision Trees, Random Forests and XGBoosted classifiers), under the assumption that the uncertainty of the input test samples can be described by certain statistical distributions. This is possible because the decision boundaries of tree-based classifiers are explicitly given (in contrast to, e.g., neural network classifiers). We extract the decision rule of each decision node of a tree-based classifier to separate the input feature space into non-overlapping regions. We then determine the probability that a random data sample wrt. the given uncertainty, which is modelled as a probability distribution, around a test sample lies in a region that has the same label as the prediction of the test sample itself. The method returns the exact probability that a data sample with distortion returns its predicted label, which is preferable in settings where exact results are crucial (e.g., safety critical applications or medical applications), whereas the measure from Scher & Trügler (2022) returns an approximation. For tree-based classifiers, our method is thus preferable, as the results are guaranteed to be exact. The method offers a tool to see how likely the returned result is given the underlying uncertainty of the input. We refrain from using the term real-world-robustness, and instead speak of probabilistic robustness.

The paper is structured as follows. First, we review various measures and concepts of robustness in Section 2. Next we describe how to quantify probabilistic robustness against natural distortions in the input for single Decision Trees in Section 3. Then the approach is extended to Random Forest classifiers and XGBoosted classifiers in Section 4. In Section 5, we present experimental results and Section 6 concludes the paper.

## 2 Related Work

One extensively studied topic is adversarial robustness (Szegedy et al., 2014), which deals with small particular manipulations of the input to cause misclassifications. These systematic manipulations are called adversarial attacks, and several algorithms have been developed (Chen et al., 2019b; Pawelczyk et al., 2020; Sharma et al., 2020) to find the nearest counterfactual (Kment, 2006; Wachter et al., 2018; Pawelczyk et al., 2021) (closest point to the input according to a distance metric that leads to misclassification) of data samples in various scenarios. Especially the area of adversarial attacks on images is highly researched since it can cause a variety of safety concerns, e.g., in medical image processing and classification (Ma et al., 2021; Kaviani et al., 2022). Zhao et al. (2018) argue that adversarial attacks are unnatural and not meaningful since the adversarial data samples are very unlikely to occur in real-world applications. They therefore developed a method to generate natural adversarial examples (Hendrycks et al., 2021) with generative adversarial networks (Goodfellow et al., 2016). Adversarial examples are found in the latent space and mapped back into the original feature space. The distance from the input data sample to the generated adversarial data samples is measured in the latent space, not in the original feature space. Pedraza et al. (2022) go a step further and argue that data samples that are generated with adversarial attacks shall not be called natural adversarial examples. In their interpretation, natural adversarial examples occur in the real-world and lead to a misclassification "without an evident cause", i.e., are caused by natural noise (e.g., from cameras or by natural changes of the input). Vasiljevic et al. (2016) investigated the robustness of classifiers against blurs in images. Hendrycks & Dietterich (2019) create benchmarks for the robustness of image classifiers with several robustness metrics on corrupted images (ImageNet-C). These alterations to the images are being referred to as common corruptions (e.g., noise and blurring in images), opposed to adversarial attacks and can be interpreted as natural adversarial examples. Hendrycks & Dietterich (2019) also define the term "corruption-robustness", which measures the rate of misclassification in test sets with included errors, but does not give a conditional probability of misclassification for single test samples as the definition of probabilistic robustness that we use. The robustness of trained classifiers is measured by evaluating the performance on unseen test data and the computation of the corruption error and variations thereof. Rusak et al. (2020) use a robustness measure that estimates the distance to the decision boundary

for each test sample, but for each sample in a random direction, and is therefore only suitable as a measure for the robustness over the whole dataset. In adversarial examples, robustness is usually measured by some distance metric, e.g., the Euclidean distance, and the distance to the closest counterfactual is used as the robustness metric. With this interpretation of robustness, one single adversarial example in the close vicinity of a test sample can have a huge influence in the evaluation of classifiers (see Scher & Trügler (2022) for a visual illustration).

Cohen et al. (2019) investigate robustness of smoothed classifiers with Gaussian noise, which come from a base classifier. A Monte-Carlo sampling approach is used to determine the most probable outcome of a data sample under the Gaussian noise. Even though the probability for each class can be computed, it is not used as the measure of robustness.

A different definition of robustness was introduced in Scher & Trügler (2022) which they termed *real-world-robustness*. Here we refrain from using the term real-world-robustness, and instead speak of probabilistic robustness. The measure describes a general framework to compute the robustness of individual predictions of a trained machine learning model against natural distortions, e.g., data-processing errors, noise or measurement errors in the input data, opposed to individual adversarial samples. The real-world-robustness $\mathbf{R}_\mu$ of the prediction $f$ of an $N$-dimensional data sample $\mu$ with distortion $p_\mu(\vec{x})$ given as a probability density function (PDF) is defined via a binary function $f'$,

$$f'(\mu, \epsilon) = \begin{cases} 0, & f(\mu + \epsilon) = f(\mu) \\ 1, & f(\mu + \epsilon) \neq f(\mu) \end{cases}$$

with $\epsilon \sim p_\mu(\vec{x})$ and an integral that determines the probability $P$ for a different prediction compared to the data sample $\mu$,

$$P(f(\mu + \epsilon) \neq f(\mu)) = \int_{\mathbb{R}^N} f'(\mu, \epsilon) \, d\epsilon.$$

The real-world-robustness $\mathbf{R}_\mu$ of the data sample $\mu$ with distortion $\epsilon \sim p_\mu(\vec{x})$ is then computed by

$$\mathbf{R}_\mu = 1 - P.$$

In words, real-world-robustness is the probability that the prediction (classification) of an input sample does not change under the given uncertainty of the input sample. Scher & Trügler (2022) showed how this probabilistic robustness measure can approximately be computed for any black-box classifier with a Monte-Carlo based method, under the constraint that the input feature space is not too high dimensional. They additionally provide a detailed discussion for the justification of this definition of real-world-robustness, and how it differs from adversarial robustness. This definition of robustness clearly distinguishes the measure from corruption-robustness, which measures the rate of misclassification in test sets with included errors, but does not give a conditional probability of misclassification for single test samples. The practical relevance of this robustness measure are settings in which the uncertainty of input samples is known (or at least approximately known), and in which one needs a measure of how likely it is that a misclassification will occur due to the random error in the input data. This could for example be applications that use multivariate sensor data with measurement noise (e.g., temperature measurements at different locations).

Other approaches for the Approaches for the quantification of robustness of trained Machine Learning models, mainly neural networks, exist. Weng et al. (2019) developed a probabilistic robustness verification tool. Their proposed metric is based on attack scenarios within $l_p$-balls around a data sample where the goal is to find the closest counterfactual of a specified target class, called the certified lower bound. To relax the condition of the closest counterfactual, the idea is to compute the largest distance between an input sample and the closest counterfactual that can be certified with a certain confidence. Mangal et al. (2019) introduce the term probabilistic robustness of neural networks. A neural network is probabilistically robust if for any two input data samples drawn form the input probability distribution, the probability that the distance between the outputs is smaller than the distance between the inputs multiplied by a constant, given that the inputs are

not too far apart, is larger than a predefined threshold. These definitions of probabilistic robustness differ from the notion of real-world-robustness introduced in Scher & Trügler (2022), which allows samples from the entire feature space as inputs, whereas Weng et al. (2019) and Mangal et al. (2019) only consider restricted inputs. Cohen et al. (2019) investigate robustness of smoothed classifiers with Gaussian noise, which come from a base classifier. A Monte-Carlo sampling approach is used to determine the most probable outcome of a data sample under the Gaussian noise. Even though the probability for each class can be computed, it is not used as the measure of robustness.

Another research direction deals with robust adversarial training of machine learning models. Qian et al. (2022) present a comprehensive survey on robust adversarial training by introducing the fundamentals and a general theoretical framework, and by summarising different training methodologies against various attack scenarios. Tan et al. (2022) introduce a training framework by adding an adversarial sample detection network to improve the classifier. In tree-based models, a training framework to learn robust trees against adversarial attacks has been developed by Chen et al. (2019a), and Ghosh et al. (2017) investigate the robustness of Decision Trees with symmetric label noise in the training data. Chen et al. (2019b) propose a robustness verification algorithm for tree-based models to find the minimal distortion in the input that leads to a misclassification. Rusak et al. (2020) propose a training algorithm that makes image classifiers more robust against naturally occurring noise.

In this paper, we quantify probabilistic robustness by exactly computing the measure from (Scher & Trügler, 2022) for tree-based classifiers (Decision Trees, Random Forests and XGBoosted classifiers), under the assumption that the uncertainty of the input test samples can be described by certain statistical distributions. This is possible because the decision boundaries of tree-based classifiers are explicitly given (in contrast to, e.g., neural network classifiers). We extract the decision rule of each decision node of a tree-based classifier to separate the input feature space into non-overlapping regions. We then determine the probability that a random data sample wrt. the given uncertainty, which is modelled as a probability distribution, around a test sample lies in a region that has the same label as the prediction of the test sample itself. Our measure of robustness is a probability which differs from distance metrics that are commonly used for adversarial robustness.

The paper is structured as follows. First, we describe how to quantify probabilistic robustness against natural distortions in the input for single Decision Trees in Section 3. Then the approach is extended to Random Forest classifiers and XGBoosted classifiers in Section 4. In Section 5, we present experimental results and Section 6 concludes the paper.

## 3 Robustness of Decision Trees

At first we show how to quantify probabilistic robustness against natural distortions in the input for a trained Decision Tree (DT) classifier (Quinlan, 1986) with a categorical target variable. We extract the decision rule from each decision node of a trained DT to split the input feature space into non-overlapping regions. For two-dimensional inputs, the regions are rectangles and for higher dimensions, the regions are hyperrectangles. For ease of description, we call the regions *boxes* (Chen et al., 2019b). To determine the robustness of the prediction of a data sample with uncertainty (e.g., noise or measurement errors), we classify the data sample with the trained DT and compute the probability that a random sample wrt. the given uncertainty is in a box that has the same label as the data sample. Taking the sum over the computed probabilities returns the robustness of the DT classification for that particular data sample.

### 3.1 Segmentation of the feature space

We have a trained DT without prior knowledge about the input features $X_i$. Each decision node in a DT is a decision rule of the form $X_i \leq \tau_{ij}$, where $\tau_{ij}$ marks the $j^{th}$ decision rule of feature $X_i$. Note that $\tau_{ij}$ is unique for each $j$ in a DT. We extract the decision rule from each decision node in the tree, add it to the decision rule set $\tau_i$ of the associated feature $X_i$ and sort each $\tau_i$ in ascending order. The elements of $\tau_i$ split one dimension of the input feature space into non-overlapping segments and the individual elements of two sets $\tau_j$ and $\tau_k$ are orthogonal to each other. Using two successive elements of each decision rule set $\tau_i$ to split

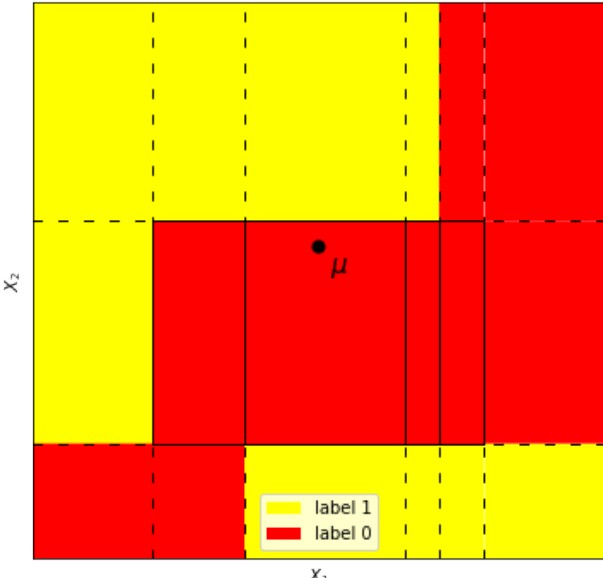

Figure 1: Illustration of boxes of a trained binary Decision Tree with two input features $X_1, X_2$ and a data sample $\mu$.

the input feature space creates one individual box. If a feature $X_i$ is bounded, we expand $\tau_i$ to its minimum and/or maximum values, otherwise we expand $\tau_i$ to negative and positive infinity to cover the entire input feature space, i.e., if $\tau_i = \{60, 80, 100\}$, the expanded unbounded set is $\tau_i' = \{-\infty, 60, 80, 100, \infty\}$. This results in a total number of boxes $n_b$ given by

$$n_b = \prod_i^N (|\tau_i'| - 1) = \prod_i^N (|\tau_i| + 1),\tag{3}$$

where $N$ is the number of input features and $|\tau_i|$ is the number of decision rules for feature $X_i$.

## 3.2 Quantifying robustness

We use the created boxes to quantify the robustness $\mathbf{R}_\mu$ of the prediction of an input data sample $\mu$ with associated uncertainty $p_\mu(\vec{x})$. We determine the predicted label of $\mu$ as well as the labels of all boxes by classifying their centre with the DT. To compute the robustness of the prediction of $\mu$, it suffices to only consider the boxes that have the same label as $\mu$ itself, denoted as $\mathbf{B}_\mu$, since data samples in these boxes return the same result as $\mu$. We determine the probability mass $m_B$ that each box $B \in \mathbf{B}_\mu$ is covering wrt. the given uncertainty $p_\mu$ around the data sample $\mu$. Taking the sum over the determined probability masses returns the robustness of the prediction of $\mu$. Figure 1 illustrates the classified boxes (two labels) of a trained DT with two input features $X_1, X_2$ and a data sample $\mu$.

In case the uncertainty distribution of the data sample $\mu$ is analytically tractable (e.g., a multivariate normal distribution with uncertainty $\Sigma$), an exact quantification of the probabilistic robustness can be computed. We determine the probability mass $m_B$ that each box $B \in \mathbf{B}_\mu$ is covering by integrating the probability density function $p_\mu(\vec{x})$ of the underlying uncertainty distribution between the lower and upper boundaries of each box. Taking the sum over all computed probability masses gives the robustness $\mathbf{R}_\mu$ of the prediction of the data sample $\mu$,

$$\mathbf{R}_\mu = \sum_{B \in \mathbf{B}_\mu} \int \cdots \int_{B_{\text{low}}}^{B_{\text{upp}}} p_\mu(\vec{x}) \, dx_1 dx_2 \ldots dx_N,\tag{4}$$

where $B_{\text{low}}$ denotes the lower boundaries and $B_{\text{upp}}$ denotes the upper boundaries of a box.

**One-dimensional feature space**  For one-dimensional data samples $\mu$ with uncertainty given as a probability distribution, we can just integrate the PDF of the uncertainty probability distribution between the lower and upper boundary of each box $B \in \mathbf{B}_\mu$ if it is analytically tractable and take their sum to get the robustness of the prediction of $\mu$.

**Input data with multivariate normal uncertainty**  For an $N$-dimensional data sample $\mu$ with multivariate normal uncertainty $\Sigma$, we integrate the multivariate normal probability density function

$$p_\mu(\vec{x}) = \frac{1}{\sqrt{(2\pi)^N |\Sigma|}} \exp\left(-\frac{1}{2}(\vec{x} - \mu)^T \Sigma^{-1}(\vec{x} - \mu)\right) \tag{5}$$

with the method by Genz (1992) between the lower and upper boundaries of each box $B \in \mathbf{B}_\mu$ and take the sum over the probability masses, which returns the probabilistic robustness of the prediction of $\mu$.

**Non-normal multivariate uncertainty distributions**  For an $N$-dimensional data sample $\mu$ with correlated features and the uncertainty in different dimensions given by different continuous distributions, computing the probabilistic robustness via analytical integration is not always possible, since integrating the joint PDF might not be analytically tractable - except if the variables are all uncorrelated to each other. However, our method can be adapted to any multivariate probability distribution that can be transformed to a multivariate normal distribution. This is the class of probability distributions with $N$ features, that can be described by a $N \times N$ covariance matrix and a set of $N$ transformations. These probability distributions can be generated from a multivariate normal distribution (characterised by an $N$-dimensional 0-mean vector and the $N \times N$ covariance matrix) and then applying the transformations for each feature. This process is known as Normal to Anything *NORTA* (Cario & Nelson, 1997). If such a joint probability distribution is given, together with the rank correlations (e.g., Spearman's $\rho$) between the features, it can be transformed to a multivariate normal distribution in a two step procedure, which in turn can be used for our robustness calculation. The rank correlations between the features are used, since the various features might not be linearly correlated. In the first step, we apply the cumulative distribution function (CDF) of the respective distributions that model the uncertainty w.r.t. the data sample $\mu$ on the decision boundaries in each dimension, which transforms them into the $[0,1]^N$-space. Note that the rank correlations between the features remain unchanged in this step. In the second step, we use the inversion method (Devroye, 1986) by applying the inverse of the CDF of the standard normal distribution on each transformed decision boundary in the $[0,1]^N$-space. Note that the rank correlations between the features still remain unchanged. We can now transform the rank correlations $\rho_k$ between the features into linear correlation coefficients with $R_k = 2 \cdot \sin(\rho_k \cdot \pi/6)$ (Pearson, 1907), and solve Equation 4 by integrating the multivariate normal probability density function (Equation 5) with $\mu = \vec{0}$ and the covariance matrix $\Sigma$ with Ones on the main diagonal elements and the transformed linear correlation coefficients $R_k$ between the features on the off-diagonal elements between the transformed lower and upper boundaries of each box $B \in \mathbf{B}_\mu$ to compute the robustness of the prediction of a data sample $\mu$.

For an $N$-dimensional data sample with independent features, the probabilistic robustness can be computed directly via analytical integration, if the PDF of the uncertainty distribution in each dimension is analytically tractable. We determine the covered probability mass of each box $B \in \mathbf{B}_\mu$ by taking the product of the covered probability masses in each dimension and then take the sum over the probability masses $m_B$ per box.

### 3.2.1   Runtime Improvement

Computing the robustness of low-dimensional inputs and shallow trees is fast since the number of boxes is small. For high-dimensional inputs and deep trees, the number of boxes (see Equation 3) and simultaneously the runtime increases with each input dimension. One approach to decrease the runtime for quantifying the robustness of the prediction of a data sample $\mu$ with multivariate normal uncertainty $\Sigma$ is to only consider the boxes that are inside of or intersect with the bounding box of, e.g., the 99% confidence hyperellipsoid

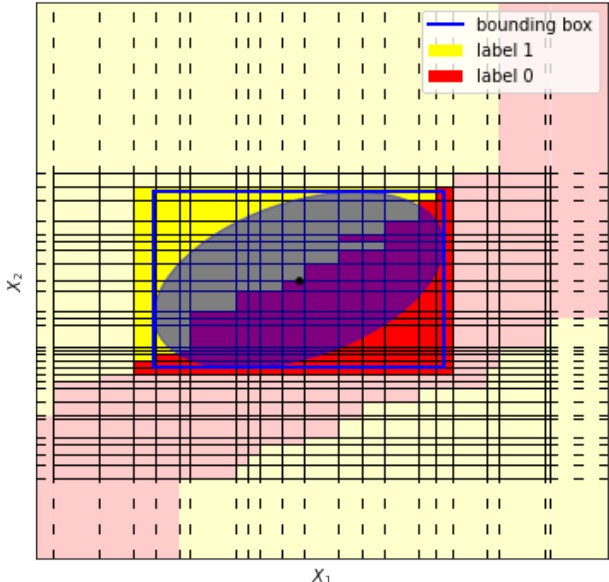

Figure 2: Boxes inside of or intersecting with the bounding box of the 99% confidence ellipse around a data sample with multivariate normal uncertainty, and more transparent boxes outside of the bounding box.

around a data sample $\mu$. Experiments have shown that the resulting robustness computations are much faster since the probability mass for less boxes needs to be computed, while the difference in the results is negligible, see Section 5. There is also a theoretical upper bound for the error of the resulting robustness (at max 1 percentage point), i.e., if the probabilistic robustness of a data sample $\mu = 0.67$, the robustness when only considering the boxes that are inside of or intersect with the bounding box of the 99% confidence hyperellipsoid around the data sample $\mu$ is in the interval $[0.66, 0.68]$. This is different to the random sampling method employed in Scher & Trügler (2022), for which a confidence interval could be computed based on concentration inequalities, but no upper or lower bound probabilistic uncertainty. Another approach to speed up computations, which has not been tested, would be to compute the probability mass $m_B$ of each box with the same label as $\mu$ in parallel.

Figure 2 represents a two-dimensional data sample $\mu$ with multivariate normal uncertainty, where the boxes that are outside of the 99% confidence ellipse are more transparent. Only using the boxes that are inside of or intersect with the bounding box of the 99% confidence ellipse to determine the robustness of the prediction of $\mu$ speeds up computations, while the difference in the results is negligible (see Section 5).

## 4  Robustness of other tree-based methods

We now extend the approach to determine the robustness of a DT to more advanced tree-based methods. We look at Random Forests, which consist of multiple trees, and at XGboosted trees.

### 4.1  Random Forest

A Random Forest (RF) (Breiman, 2001) extends a Decision Tree model and consists of multiple DTs. Each tree is trained on a bootstrapped dataset with the same size as the original training dataset, i.e., a sample can be part of the training set for one tree multiple times, whereas other samples are not part of that training set. We extract the decision rule of each decision node in all individual trees, merge them per input feature $X_i$ to form the decision rule set $\tau_i$ and create boxes to compute the robustness of the prediction of a data sample.

Analogous to the DT setup, we have a trained RF without prior knowledge about the input features $X_i$ or the individual trees. For each tree in the RF, we extract the decision rule $X_i \leq \tau_{ij}$ of each decision node, add it to the decision rule set $\tau_i$ of the associated feature $X_i$ and sort each $\tau_i$ in ascending order, as was done for DTs. Since a RF consists of multiple DTs, decision nodes in different trees can have the same decision rule $X_i \leq \tau_{ij}$, leading to duplicate entries in $\tau_i$. We eliminate all but one of the duplicate entries $\tau_{ij}$, such that each $\tau_{ij}$ is unique. To create boxes for robustness computations, we use the method described in Section 3.1 for DTs.

After creating the boxes, we compute the robustness of the prediction of a data sample $\mu$ in a RF, analogous to the approach for DTs described in Section 3.2. First we determine the label of each box by classifying its centre with the RF, not with the single trees. Then we classify $\mu$ with the RF to determine its label and take the sum over all probability masses $m_B$ of the boxes $B \in \mathbf{B}_\mu$ (boxes with the same label as $\mu$).

Computing the robustness in a RF is computationally more expensive than in a DT, since a RF consists of multiple trees, leading to more decision rules and a higher number of boxes (see Equation 3). In Section 3.2.1, we described an approach to approximate the robustness of the prediction of a data sample with multivariate normal uncertainty in a DT by only considering the boxes that are inside of or intersect with the bounding box of the 99% confidence hyperellipsoid. Experiments with various sizes of RFs (number of trees and depth of trees) have shown that computing the robustness of the prediction of a data sample is much faster when only considering these boxes, while the robustness results only differ marginally (see Section 5).

Figure 3 contains the boxes of a trained RF (Figure 3d) with two input features $X_1, X_2$, two labels and three associated DTs (Figures 3a, 3b, 3c), as well as a data sample with multivariate normal uncertainty. We see that each individual DT is represented by different decision boundaries and different boxes. Overlaying the individual Figures of the DTs gives the representation for the trained RF (Figure 3d). We also observe that the data sample is classified differently in $DT_1$ (red label) compared to the other two DTs and the RF (yellow label).

At first glance, it would actually seem simpler to compute the probabilistic robustness against natural distortions in the input of a RF by first computing it individually for each DT of the forest, and then combining the results. This is, however, not possible, as averaging the robustness of the individual trees cannot account for interdependencies (e.g., even if tree A and tree B have the same robustness, the contributions from different parts of the feature space might be different, and averaging them would lead to false results).

### 4.2 XGBoosted Decision Tree model

Similar to a Random Forest, an XGBoosted Decision Tree model (Chen & Guestrin, 2016) also consists of multiple trees. Instead of training the trees from bootstrapped datasets, the individual trees build on each other. We again extract the decision rules of each decision node in all trees and combine them per feature $X_i$ to form the associated decision rule set $\tau_i$. We create boxes which are used to compute the robustness of the prediction of a data sample. Since trees in an XGBoosted Decision Tree model build on each other, there are less different decision rules than in a RF, when the number of trees and depth of the trees is the same. This leads to a smaller number of boxes and thus computations are faster.

## 5 Experimental results

In this Section we summarise experimental results for quantifying probabilistic robustness against natural distortions in the input for tree-based classifiers. Experiments were carried out on 1 core of an Intel(R) Xeon(R) 6248 CPU @ 2.50GHz processor with 256GB RAM. We are using ~~2~~ 3 datasets for the experiments, the Iris flower dataset (Anderson, 1936)~~and~~, the MNIST dataset (Deng, 2012) and the Pima-Indians-Diabetes dataset (Smith et al., 1988). For illustration purposes, the uncertainty distribution in most experiments is prescribed as a multivariate normal distribution since we can compute solutions directly. We have not made comparisons to other notions of robustness since most other robustness techniques (e.g., adversarial examples) use a distance metric between a data sample and counterfactuals or measure the robustness of the classifier, whereas our method returns a probability ~~.~~ for each individual test sample. Scher & Trügler (2022) already compared probabilistic robustness with the distance to the closest counterfactual on two datasets, showing

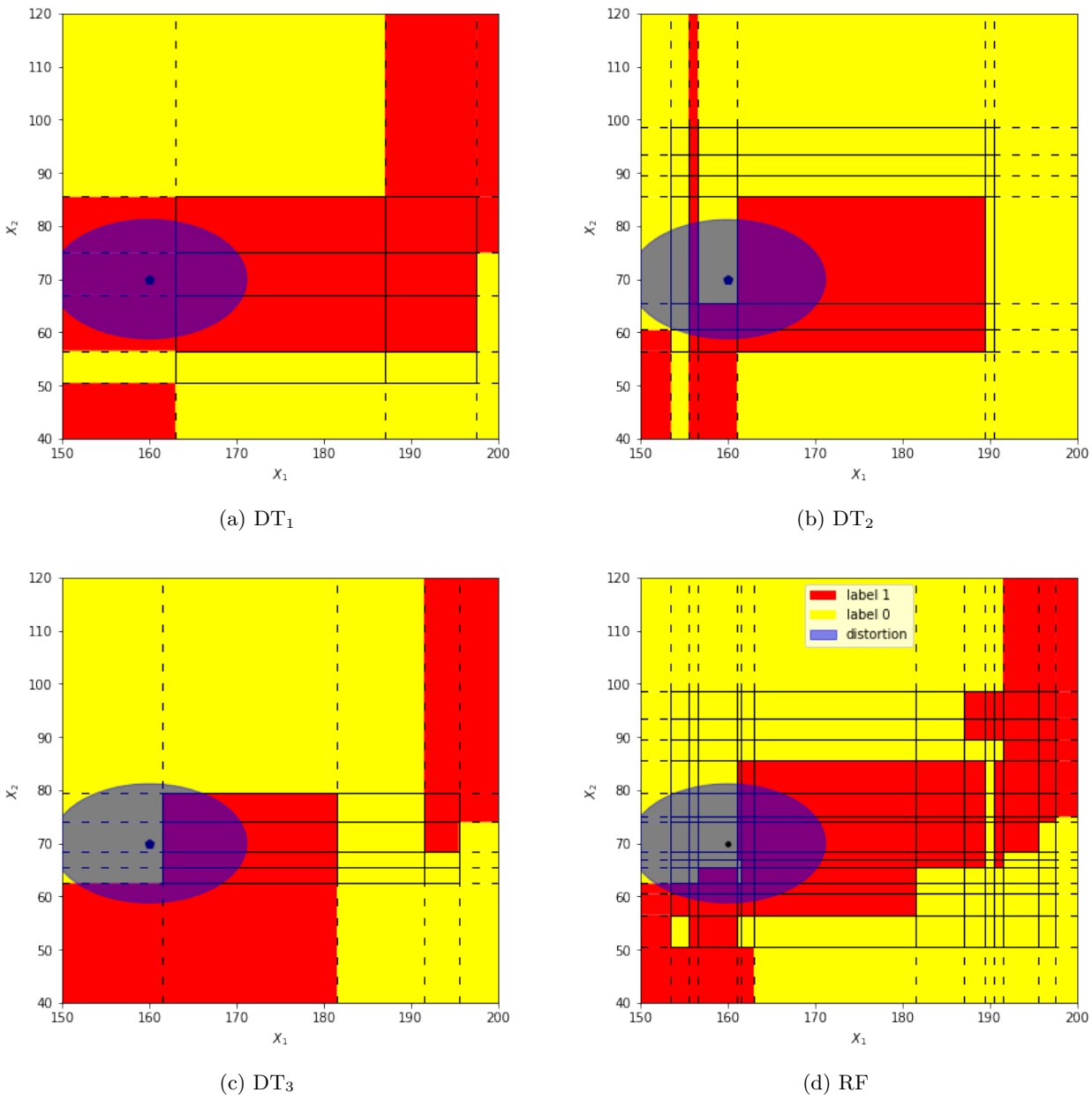

(a) DT$_1$

(b) DT$_2$

(c) DT$_3$

(d) RF

Figure 3: Boxes of 3 Decision Trees with two-dimensional input (a) - (c) and the resulting combined boxes (d) of the associated binary Random Forest.

Table 1: Comparison of probabilistic robustness results for 15 test samples of the Iris dataset using all boxes and only the boxes that are inside of or intersect with the bounding box of the 99% confidence hyperellipsoid around a test sample for robustness computation

| | All boxes | 99% boxes |
|---|---|---|
| Test sample | Boxes - Robustness | Boxes - Robustness |
| 1 | 11 - 0.77741 | 2 - 0.77740 |
| 2 | 8 - 0.99986 | 4 - 0.99985 |
| 3 | 13 - 0.94940 | 13 - 0.94940 |
| 4 | 13 - 0.98266 | 13 - 0.98266 |
| 5 | 11 - 0.99248 | 11 - 0.99248 |
| 6 | 8 - 0.99986 | 2 - 0.99985 |
| 7 | 11 - 0.77265 | 11 - 0.77265 |
| 8 | 13 - 0.58500 | 13 - 0.58500 |
| 9 | 11 - 0.56719 | 11 - 0.56719 |
| 10 | 11 - 0.85682 | 11 - 0.85682 |
| 11 | 11 - 0.99248 | 11 - 0.99248 |
| 12 | 8 - 0.99955 | 2 - 0.99955 |
| 13 | 13 - 0.98537 | 5 - 0.98537 |
| 14 | 11 - 0.86277 | 2 - 0.86276 |
| 15 | 8 - 0.99986 | 1 - 0.99985 |

that the measures are different. The repository with codes will be made available with the camera ready version.

## 5.1 Results with 99% confidence hyperellipsoid

In Section 3.2.1 we described a method to decrease the runtime of the robustness computation while only having negligible differences (at max 1 percentage point) in the results. We trained a Decision Tree with a maximum depth of 4 on the Iris flower dataset with a train/test split of 90/10, yielding 15 test samples. To compute the robustness of the prediction of the test samples, the uncertainty distribution of the test samples is given as a multivariate normal distribution with 0.1 on the main diagonal and Zeros on the off-diagonals of the covariance matrix for illustration purposes. In real applications, the uncertainty needs to be prescribed based on the exact application setting, requiring domain-knowledge. In the experiments, we determine the robustness of the prediction of the test samples computed with all boxes and the robustness, where we only look at the boxes that are inside of or intersect with the bounding box of the 99% confidence hyperellipsoid around the test samples. Table 1 lists the robustness of the 15 test samples computed with all boxes and the boxes that are inside of or intersect with the bounding box of the 99% confidence hyperellipsoid. Even though the number of boxes diverges heavily for some test samples, we see that the robustness results are very similar and only start to differ in the fifth decimal place. Figure ?? shows the probabilistic robustness results of the 15 test samples in comparison. We observe that the data points are almost on the straight dashed red line with slope 1, which illustrates the similarity between the results. We We achieved $R^2$-scores exceeding 0.9999 in each test run (equivalent to the interpretation of the $R^2$-score in a linear regression model), showing the similarity between the results and that it suffices to only consider the boxes that are inside of or intersect with the bounding box of the 99% confidence hyperellipsoid to compute the robustness of the prediction of the test samples.

Illustration of robustness results for 15 test samples of the Iris dataset using all boxes and only the boxes that are inside of or intersect with the bounding box of the 99% confidence hyperellipsoid around a test sample. The dashed red line is straight.

## 5.2 Runtime Analysis

Considering only the boxes that are inside of or intersect with the bounding box of the 99% confidence hyperellipsoid around a test sample not only returns accurate results for the probabilistic robustness of the

Table 2: Comparison of runtimes for 10 test samples of the MNIST dataset using all boxes and only the boxes that are inside of or intersect with the bounding box of the 99% confidence hyperellipsoid around a test sample for robustness computation

|  | All boxes | 99% boxes |
| Test sample | Boxes - Runtime | Boxes - Runtime |
| --- | --- | --- |
| 1 | 19,118 - 488s | 296 - 8s |
| 2 | 24,616 - 589s | 3,848 - 99s |
| 3 | 32,564 - 502s | 1,824 - 46s |
| 4 | 39,636 - 697s | 52 - 1s |
| 5 | 13,356 - 274s | 1,512 - 39s |
| 6 | 32,564 - 518s | 1,824 - 45s |
| 7 | 13,356 - 241s | 48 - 2s |
| 8 | 48,128 - 927s | 380 - 10s |
| 9 | 32,564 - 468s | 120 - 3s |
| 10 | 5,736 - 153s | 24 - 1s |

Table 3: Comparison of results for 9 trained Decision Trees on the MNIST dataset using the boxes that are inside of or intersect with the bounding box of the 99% confidence hyperellipsoid around a test sample and using the method in Scher & Trügler (2022) for probabilistic robustness computation

| Resized Image Size | DT depth | $R^2$-score |
| --- | --- | --- |
| $3 \times 3$ | 4 | 0.99990 |
| $3 \times 3$ | 5 | 0.99999 |
| $3 \times 3$ | 6 | 0.99999 |
| $5 \times 5$ | 5 | 0.99999 |
| $5 \times 5$ | 6 | 0.99998 |
| $8 \times 8$ | 4 | 0.99999 |
| $8 \times 8$ | 5 | 0.99999 |
| $10 \times 10$ | 4 | 0.99999 |
| $10 \times 10$ | 5 | 0.99998 |

prediction of a test sample, but is also much faster. With the Iris flower dataset, the runtime difference for the test samples was negligible since it only contains 4 input features. We therefore conducted experiments with the MNIST dataset. We resized the images from $28 \times 28$ to $5 \times 5$ pixels, normalised them, flattened them into a vector and used the 25-dimensional vector as input to train a Random Forest with 5 trees and a maximum depth of 3 per tree on the training set (60,000 training samples). To evaluate the runtime of the robustness computation, we determined the probabilistic robustness of the first 10 images of the test set (10,000 test samples). The uncertainty distribution of the test samples is given as a multivariate normal distribution with 0.001 on the main diagonal elements and Zeros on the off-diagonal elements of the covariance matrix for illustration purposes. The resulting runtimes and number of boxes for the robustness computation of the 10 test samples are listed in Table 2. We see that the number of boxes that are inside of or intersect with the bounding box of the 99% confidence hyperellipsoid is much smaller and that the runtimes are much shorter, compared to using all boxes. The difference in the results is negligible as we again achieved $R^2$-scores exceeding 0.9999, further indicating that it suffices to only use the boxes that are inside of or intersect with the bounding box of the 99% confidence hyperellipsoid to compute the probabilistic robustness of the prediction of test samples. We additionally computed results with the approximate method introduced in Scher & Trügler (2022) to compare the runtimes. Randomly sampling 1 million points and computing the robustness results takes approximately 3 seconds per test point. Increasing the number of randomly sampled data points by a factor of 10 also increases the runtime by a factor of 10.

## 5.3  Comparison to approximate probabilistic robustness computation

Scher & Trügler (2022) introduced the term real-world-robustness, but their method is based on random sampling and therefore only returns approximate results. With this comes the limitation that it only works

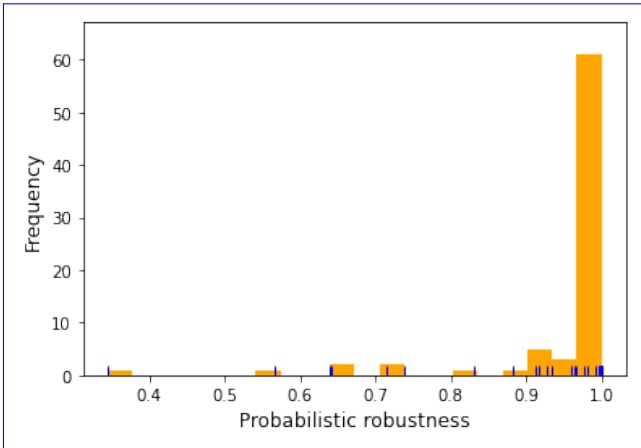

Figure 4: ~~Illustration~~ Distribution of probabilistic robustness ~~results~~ for ~~15~~ 73 test samples of an XGBoost classifier trained on the ~~Iris~~ Pima-Indians-Diabetes dataset ~~for correlated features with mixed distortions,~~ uncertainty of datapoints estimated via an assumed signal-to-noise ratio of 10/1.

~~when the feature dimensionality is not too high.~~ We compare the two methods (Random Sampling (sampling 1,000,000 times) against 99% confidence hyperellipsoid around a test sample) by computing the probabilistic robustness of the prediction of data samples on trained Decision Trees. The DTs have a depth between 4 and 6 and were trained on the MNIST dataset. We perform the same data preprocessing steps as for the runtime analysis, but resize the images to various sizes, ranging from $3 \times 3$ to $10 \times 10$ such that the DTs have input feature dimensions ranging from 9 to 100, and train them on the training set. The uncertainty is given as a multivariate normal distribution with 0.0001 on the main diagonal elements and Zeros on the off-diagonal elements of the covariance matrix for illustration purposes. We compare the results of each setting for the first 10 test samples and observe that both methods return similar results for the robustness of the prediction of data samples, with negligible differences starting in the third decimal place and $R^2$-score exceeding 0.9999 for all settings, see Table 3 for a summary of the results. This indicates that the method of Scher & Trügler (2022) to compute the robustness still works well with an input feature dimension of 100, and more input features would be necessary to see a difference in the results. We also computed the robustness of data samples with higher values in the main diagonal of the multivariate uncertainty distribution (0.001, 0.01, 0.1 and 0.5) to cover more parts of the input feature space and compared the results, but again only observed minor differences.

## 5.4 Robustness computation for correlated features and mixed distortions

In case the features are correlated and the uncertainty distribution of data samples is best modelled by different probability distributions in different dimensions, we can compute the robustness of the prediction of a data sample $\mu$ by utilising the reversed *NORTA* principle presented in Section 3.2. To illustrate the approach, we trained an XGBoosted Decision Tree model with 4 trees and a maximum depth of ~~4 and 6 trees on the Iris flower dataset~~ 3 on the Pima-Indians-Diabetes dataset (Smith et al., 1988) with a train/test split of 90/~~10. For illustration purposes the uncertainty of the 4 features is given by different distributions - normal for feature 1 (sepal length), exponential for feature 2 (sepal width), $\chi^2$ for feature 3 (petal length) and lognormal for feature 4 (petal width) - with different parameters,~~ 10 and computed the robustness of the test samples. At first we removed subjects that had a value of 0 for one of the variables "Glucose", "BloodPressure" or "BMI", and ~~after transforming the data, the correlation matrix is given with Ones on the main diagonal and values between 0.1 and 0.3 on the off-diagonals. The robustness results of the 15 test samples are visualised in Figure ?? and we observe that several test samples have a robustness of exactly 1. This is caused by uncertainty distributions with a lower bound (e.g., exponential or $\chi^2$ distribution). In case the value of a test sample in a dimension that is modelled by such a probability distribution is higher than the highest value for the decision boundaries in that dimension, the data point in this dimension will always~~

~~be above the highest value of the decision boundary. For a visual explanation, if the data sample $\mu$ in Figure 1 was in the upper right box (or a neighbouring box thereof) and the uncertainty in both dimensions was best modelled by two exponential distributions, the data sample would have a robustness of~~ removed the variable "SkinThickness" since it only had a minor influence on the performance of the resulting classifier, leading to 7 input features and 724 subjects. To estimate the uncertainty of the data points, domain knowledge about the measuring instruments (e.g., for Glucose) is necessary. Since we do not have access to the specific measuring devices and their uncertainty, we estimate the uncertainty of the data points based on the distributions of the input features. We therefore estimated the underlying distributions of the 7 input features based on histograms which suggest that the variables Glucose, BloodPressure, BMI and DiabetesPedigreeFunction seem to follow a normal distribution whereas the variables Pregnancies, Insulin and Age seem to rather follow an exponential distribution. We use these distributions to describe the uncertainty of the input features of a data point by prescribing a signal-noise ratio factor of 1~~since no smaller values could be sampled~~/10 to downscale the variance of the variables. Next we computed the rank correlations between the variables and trained an XGBoosted Decision Tree which returned an accuracy of 75% on the test samples. The robustness of the 73 test samples ranges from 0.34 to 1.00 and results are plotted in Figure 4. 55 test samples have a probabilistic robustness above 0.99, which gives an indication of the certainty of the results for these samples. The rather small probabilistic robustness of 0.34 for one test sample suggests that this result is uncertain and a small change in one input feature would lead to a different result.

## 6 Conclusion

In this paper, we presented a method for quantifying the probabilistic robustness against natural distortions in the input for tree-based classifiers. We extract the decision rules from a trained classifier to separate the input feature space into non-overlapping regions, called boxes, and integrate the underlying probability distribution that models the uncertainty of a test sample between the lower and upper boundaries of each box to determine their covered probability mass. Taking the probability sum over the boxes that have the same label as the test sample returns the robustness of the prediction of the test sample. We presented this approach for Decision Trees in detail and discussed the extension to Random Forests and XGBoosted trees. The method quantifies the probabilistic robustness of the prediction of individual data samples for tree-based classifiers. This differs from other robustness measures which use various techniques to look for adversarial examples and use the distance between the test sample and the closest counterfactual as the measure for robustness. It also differs from robustness to common corruptions, which measures the average rate of misclassification of a test set with perturbed samples, compared to our approach, which measures the uncertainty of individual test samples.

One limitation of our approach is that the uncertainty distribution needs to be modelled as a continuous probability distribution and that the PDF of the distribution must be analytically tractable to compute the probabilistic robustness. This is not always possible, especially when features are correlated and the uncertainty in different dimensions is best modelled by different distributions. Our method works for all $N$-dimensional continuous multivariate distributions that follow the *NORTA* principle, thus be described by a $N \times N$ covariance matrix and a set of associated $N$ transformations. While there are specific types of uncertainty where this is not possible (e.g. salt-and-pepper noise in computer vision), in many tasks, especially those that use tabular data measured by physical devices, the uncertainty will - at least in most cases - follow an uncertainty that can be transformed into a multivariate Gaussian with the *NORTA* principle In case the uncertainty distribution is more complex (e.g., because it is not given as a probability distribution, but by a stochastic function that models some process), we can in principle use numerical integration techniques to integrate the probability mass over the boxes for classifiers with explicit decision boundaries (such as tree-based models) and find approximate solutions. We carried out some initial experiments in that direction, solving Equation 4 with the *quadpy* package (Schlömer et al., 2021) for numerical integration. We observed however that the results with numerical integration techniques are often unstable and unreliable, especially when a data sample $\mu$ is in a large box, i.e., there is a big gap between the lower and upper boundaries in at least one dimension of the box, compared to cases that are actually analytically solvable. While it could be possible that with more carefully choosing types and parameters of the numerical routines

the results could be better, this shows that simply using off-the-shelf integration routines is not a feasible option.

A second limitation of our approach is the high amount of data storage space needed. Equation 3 shows the number of boxes that are being created for a trained classifier. Classifiers with high input feature dimension and especially Random Forests quickly exceed the available storage space, such that the robustness cannot be computed anymore. With a more powerful machine, experiments on tree-based classifiers with high input feature dimension could be carried.

The method presented in this paper is applicable because tree-based classifiers have the convenient property of having explicitly described decision boundaries that form hyperrectangles. Future research should be dedicated to extending the presented approach to more advanced classifiers with complicated decision boundaries, such as (nonlinear) Support Vector Machines and Neural Networks.

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
