# OpenReview forum: "Quantifying probabilistic robustness of tree-based classifiers against natural distortions"
_TMLR — Rejected by TMLR_

### Review · Reviewer_6sLD · 2022-12-16

**Summary Of Contributions:**

The authors develop a method for measuring the robustness of decision tree-based classifiers to certain families of input noise distributions. The core idea is that the input space of a decision tree can be partitioned into regions, each of which corresponds to a fixed label prediction. Then, given an input and a noise distribution, the authors directly compute the probability that the noise will cause the input to end up in a region with a label that is different from the original prediction.

To make their method more efficient, the authors do not evaluate input-space regions which are too far from the input (they evaluate regions which overlap with the ellipsoid in which the noisy input will end up with probability 99%). The authors extend their method to ensembles of decision trees (XGBoost).

**Audience:**

No

**Claims And Evidence:**

No

**Requested Changes:**

(none of the changes suggested would have a significant impact on my evaluation of the paper)

Questions worth clarifying:

- Does the runtime for the “99% confidence hyperellipsoid” include finding the relevant boxes?
- How does the runtime of the approximate method compare to the exact one?
- For the approximate method, why is the number of dimensions important? If the goal is to estimate model robustness through random noise samples, we are effectively estimating a scalar random variable through independent samples, which, through standard concentration, should converge fairly quickly to the true mean.

Writing improvements:

- Abstract: I would put the term “real-world-robustness” in quotes.
- Introduction:
    - The discussion around adversarial examples is too lengthy given that these are largely orthogonal to the contributions of the paper.
    - The first paragraph is really long and mostly discussed related work as opposed to contextualizing the paper.
    - In general, the main point of the introduction should be to convey that: (a) “noise-robustness is important”, (b) “we propose a method for exactly evaluating the noise-robustness decision trees”.

**Strengths And Weaknesses:**

### strengths

Noise-robustness is an important metric for assessing the reliability of a model and the authors provide an exact method for measuring it for a class of models.

### weaknesses

The utility of the proposed method is not entirely clear. Since the goal is to measure the robustness of a model to random perturbations, a much simpler method where one repeatedly samples random perturbations would be just as effective in practice. Given that the proposed method only works for specific models and noise distributions, the authors should provide some justification as to what its advantages are.

The technical and conceptual depth of the study is rather limited. The proposed method is a direct computation of a natural quantity for which we already have effective alternatives (see above). Thus, I don’t believe that this study would be of interest to the broader TMLR audience.

Finally, the experimental validation of the method is really limited. The authors evaluate on two very simple datasets (IRIS and MNIST) using less than 20 test examples for each.

---

> ### Author Response · Authors · 2023-01-18
> **Response to Reviewer 6sLD**
>
> We thank the reviewer 6sLD for the time and the feedback, and for the opportunity to perform a revision on the submitted manuscript "Quantifying probabilistic robustness of tree-based classifiers against natural distortions". Please find below a response to the comments.
>
>
> - Finally, the experimental validation of the method is really limited. The authors evaluate on two very simple datasets (IRIS and MNIST) using less than 20 test examples for each.
>
>       We have added an experiment with a real-life dataset ("Using the ADAP learning algorithm to forecast the onset of diabetes mellitus") in Section 5.4 in the revised version of the manuscript to show a relevant application of our method. The dataset contains 768 data points with 8 variables and a binary outcome variable. We eliminated data points that contain unrealistic numbers which reduced the number of data points to 724. We further removed one variable since it only had a minor impact on the outcome, resulting in 7 input variables. Next, we computed the rank correlations between the variables and estimated the distributions of the input features based on histograms which showed that the variables seem to either follow a normal distribution or an exponential distribution. To estimate the uncertainty of the single data points, we prescribed a signal-noise ratio factor of 1/10 to scale the variance of the estimated best fitting distributions. In a real setting, the estimate of the uncertainty would have to come from information from the manufacturer of the devices used for the measurements. Finally we trained an XGBoosted classifier, where we achieved an accuracy of approximately 75% on the test samples (90/10 train/test-split) and computed the probabilistic robustness of test samples.
>
> - Does the runtime for the “99% confidence hyperellipsoid” include finding the relevant boxes?
>
>       Finding the relevant boxes is not included in the runtime. This step only takes a negligible amount of time such that we did not mention it in the manuscript.
>
> - How does the runtime of the approximate method compare to the exact one?
>
>       This depends on the number of sampled data points in the approximate method and also on the configuration of the classifier (depth of the trees, number of trees). Table 2 shows a comparison between the method with all boxes against only using boxes that are inside of or intersect with the bounding box of the 99% confidence hyperellipsoid. Randomly sampling 1 million points and computing the robustness results takes approximately 3 seconds per test point. Increasing the number of randomly sampled data points by a factor of 10 also increases the runtime by a factor of 10. We added a short analysis at the end of Section 5.2 in the revised version of the manuscript.
>
> - For the approximate method, why is the number of dimensions important? If the goal is to estimate model robustness through random noise samples, we are effectively estimating a scalar random variable through independent samples, which, through standard concentration, should converge fairly quickly to the true mean.
>
>       For high-dimensional examples the method does not guarantee that through random sampling a significant portion of the input is covered. We are unfortunately not familiar with the term "standard concentration". We checked both online and in several statistics text books, and we were not able to find usage of the term in connection to statistics.
>
> - Abstract: I would put the term “real-world-robustness” in quotes.
>
>       We thank the reviewer for the suggestion and change it in the revised version of the manuscript.
>
> - The discussion around adversarial examples is too lengthy given that these are largely orthogonal to the contributions of the paper. The first paragraph is really long and mostly discussed related work as opposed to contextualizing the paper.
> In general, the main point of the introduction should be to convey that: (a) “noise-robustness is important”, (b) “we propose a method for exactly evaluating the noise-robustness decision trees”.
>
>       We have now restructured the paper by separating the first Section into two sections (Introduction and Related Work) to improve the reading flow.

---

> > ### Comment · Reviewer_6sLD · 2023-01-20
> > **Advantage of proposed algorithm over random noise sampling still not supported**
> >
> > I appreciate the authors’ response and updates to the manuscript.
> >
> > **Concentration bounds for the approximate algorithm.** I apologize for not being more clear: When talking about concentration, I was referring to the notion of using [concentration inequalities](https://en.wikipedia.org/wiki/Concentration_inequality) to establish [confidence intervals](https://en.wikipedia.org/wiki/Confidence_interval). Specifically, by applying [Hoeffding’s inequality](https://en.wikipedia.org/wiki/Hoeffding's_inequality) directly, one can prove that estimating the robustness of *any* given test point by averaging the model’s performance on 1 million random samples from the noise distribution will yield an answer within 0.002 of the true mean with at least 99.96% probability. Note that this is independent of the number of input dimensions since it only depends on the fact that we are estimating *a single scalar* quantity (and the fact that we can sample independent noise samples).
> >
> > **Utility of exact computation.** The authors mention in the updated version of the manuscript that the approximate algorithm takes ~3s per test point (with 1M noise samples/point). This means that it is *orders of magnitude* *faster* than the exact algorithm (according to Table 2). Moreover, even using the 99%-confidence-box approximation proposed by the authors (which means that the algorithm is no longer exact), the runtime is still slower in most cases than simply using random noise to estimate the mean. Finally, given the argument above, the precision of the approximate algorithm should be sufficient for most practical purposes.
> >
> > Overall, it is still not clear to me why someone would use the proposed algorithm---which imposes restrictions to the model structure and noise distribution---over simply using random samples to estimate the mean.

---

> > > ### Author Response · Authors · 2023-01-23
> > > **Reply to "Advantage of proposed algorithm over random noise sampling still not supported"**
> > >
> > > - Concentration bounds for the approximate algorithm. I apologize for not being more clear: When talking about concentration, I was referring to the notion of using concentration inequalities to establish confidence intervals. Specifically, by applying Hoeffding’s inequality directly, one can prove that estimating the robustness of any given test point by averaging the model’s performance on 1 million random samples from the noise distribution will yield an answer within 0.002 of the true mean with at least 99.96% probability. Note that this is independent of the number of input dimensions since it only depends on the fact that we are estimating a single scalar quantity (and the fact that we can sample independent noise samples).
> > >
> > >       Thank you for clarifying your comment, and for pointing out to us that also for an approximate method based on random sampling, upper bounds for the probability that the answer will be within a certain confidence interval can be computed. This is something we were not aware of, and that we now mention in the manuscript (section 3.2.1). This is, however, a different type of upper bound than the one that our approximate algorithm yields. Our method gives an upper bound for the uncertainty of the target variable itself (equivalently we could say that it gives us a lower and an upper bound for the  target variable itself), not for the probability that the target variable is in a certain interval. The discussion around this might have been a bit confusing, as our target variable - probabilistic robustness -  is a probability itself. If we exclude all boxes outside the 99% interval of the uncertainty distribution, and the algorithm tells us that probabilistic robustness of a data sample is, e.g., 0.67, then we know for sure that the true probabilistic robustness is in the interval [0.66, 0.68]. We have made additions in Section 3.2.1 and in Section 5.1 to make this clearer to the reader.
> > >
> > > - Utility of exact computation. The authors mention in the updated version of the manuscript that the approximate algorithm takes ~3s per test point (with 1M noise samples/point). This means that it is orders of magnitude faster than the exact algorithm (according to Table 2). Moreover, even using the 99%-confidence-box approximation proposed by the authors (which means that the algorithm is no longer exact), the runtime is still slower in most cases than simply using random noise to estimate the mean. Finally, given the argument above, the precision of the approximate algorithm should be sufficient for most practical purposes.
> > > Overall, it is still not clear to me why someone would use the proposed algorithm---which imposes restrictions to the model structure and noise distribution---over simply using random samples to estimate the mean.
> > >
> > >       We agree with the reviewer that the results of the proposed method, the method when using the 99%-confidence-box and the approximate method are all similar and that an approximate method is sufficient for most practical purposes. In settings where exact results are crucial (e.g., safety critical applications or medical applications), our proposed exact method is preferable over any approximate alternative. Another way to see it is that even if the approximation of random sampling is good, if the classifier is a tree-based classifier, there is no reason to use random sampling, because our method can yield exact results. We have made a small addition in the second to last paragraph of Section 1 to point this out. Additionally we want to point out that in our manuscript we do not claim that the random sampling method is not useful, we only show that for tree-based classifiers there is also an exact way to compute probabilistic robustness. Finally, independent of the question of practical relevance, our contribution of showing how this type of robustness can be computed in an exact way is a theoretical result that is of interest already on its own. Considering the above points as well as the scope of TMLR, we are convinced that the method is of interest to at least some readers of TMLR.

---

### Review · Reviewer_tnv3 · 2022-12-17

**Summary Of Contributions:**

Contribute a method for exactly computing "probabilistic robustness", a measure introduced by (Scher & Trügler, 2022), for tree-based classifiers assuming input distortions are sampled from certain statistical distributions.

**Audience:**

Yes

**Claims And Evidence:**

Yes

**Requested Changes:**

Overall, it is unclear what insights the evaluation metric offers on problems of interest. If I am a practitioner aiming to solve a problem, how does this metric help me, relative to alternatives in related work?

If the authors have an answer to this question, it's important to provide supporting empirical evidence.

**Strengths And Weaknesses:**

Strengths:
- Clearly written; the deep dives in Sec. 2 & 3 were executed well.
- Contextualizing contribution w/ related work improved.

Weaknesses:
- Experiments demonstrate tractable computation & tests in less restrictive settings, but in terms of the method's utility, sec. 4.3 suggests the introduced method is useful for verifying the approximate approach of (Scher & Trügler, 2022), but not when this method is preferable to that of (Scher & Trügler, 2022).
- While the authors' motivate the practical relevance, they do not demonstrate it. Are there real-world problem settings which match the method's assumptions where the authors could show their method is useful?

---

> ### Author Response · Authors · 2023-01-18
> **Response to Reviewer tnv3**
>
> We thank the reviewer tnv3 for the time and the feedback, and for the opportunity to perform a revision on the submitted manuscript "Quantifying probabilistic robustness of tree-based classifiers against natural distortions". Please find below a response to the comments.
>
> - Clearly written; the deep dives in Sec. 2 & 3 were executed well.
>
>       We thank the reviewer for the positive feedback.
>
> - Experiments demonstrate tractable computation & tests in less restrictive settings, but in terms of the method's utility, sec. 4.3 suggests the introduced method is useful for verifying the approximate approach of (Scher & Trügler, 2022), but not when this method is preferable to that of (Scher & Trügler, 2022).
>
>       Our proposed method gives exact results whereas the method by (Scher & Trügler, 2022) only gives approximate results. Therefore in settings where exact results are crucial, our proposed method is preferable. We have added additional discussion on this in the second to last paragraph of the Introduction in the revised version of the manuscript.
>
> - While the authors' motivate the practical relevance, they do not demonstrate it. Are there real-world problem settings which match the method's assumptions where the authors could show their method is useful?
>
>       We have added an experiment with a real-life dataset ("Using the ADAP learning algorithm to forecast the onset of diabetes mellitus") in Section 5.4 in the revised version of the manuscript to show a relevant application of our method. The dataset contains 768 data points with 8 variables and a binary outcome variable. We eliminated data points that contain unrealistic numbers which reduced the number of data points to 724. We further removed one variable since it only had a minor impact on the outcome, resulting in 7 input variables. Next, we computed the rank correlations between the variables and estimated the distributions of the input features based on histograms which showed that the variables seem to either follow a normal distribution or an exponential distribution. To estimate the uncertainty of the single data points, we prescribed a signal-noise ratio factor of 1/10 to scale the variance of the estimated best fitting distributions. In a real setting, the estimate of the uncertainty would have to come from information from the manufacturer of the devices used for the measurements. Finally we trained an XGBoosted classifier, where we achieved an accuracy of approximately 75% on the test samples (90/10 train/test-split) and computed the probabilistic robustness of test samples.
>
>
> - Overall, it is unclear what insights the evaluation metric offers on problems of interest. If I am a practitioner aiming to solve a problem, how does this metric help me, relative to alternatives in related work? If the authors have an answer to this question, it's important to provide supporting empirical evidence.
>
>       The evaluation metric returns the probability that a data sample with distortion returns its predicted label, and it does this in a an exact way. It therefore offers a tool to see how likely the returned result is given the underlying uncertainty of the input. We have added more discussion on this at the second to last paragraph of the Introduction in the revised version of the manuscript. Please note that we have separated the former  Introduction section into two sections, Introduction and Related Work.

---

### Review · Reviewer_xKGp · 2023-01-04

**Summary Of Contributions:**

This paper proposes a method for exactly computing a notion of real-world/probabilistic robustness for tree-based classifiers.

**Audience:**

No

**Broader Impact Concerns:**

No concerns, understanding robustness is generally desirable.

**Claims And Evidence:**

Yes

**Requested Changes:**

For me to recommend acceptance, besides addressing some of the writing and experiment concerns, I would really like to see a relevant application or at least an abstraction thereof. Gaussian noise on (downscaled) MNIST or IRIS is in my opinions not interesting enough.

**Strengths And Weaknesses:**

Strengths:
- It is generally beneficial to be able to compute notions or robustness exactly.
- Simple and generalizable method that works for multiple tree-based methods.

Weaknesses:
- Regarding the first paragraph: in the adversarial robustness community, many papers actually report something like robust accuracy (accuracy on adversarial examples) instead or in addition to the distance to the adversarial example.
- The second paragraph about Cohen et al. feels a bit unconnected and misplaced.
- In equation (2), I am wondering whether there is p_\mu missing? Currently, the integral runs over R^N and is not weighted by the underlying noise distribution p_\mu. Also, why not invert f’ and define R_\mu directly as equation (2) for simplicity?
- The authors state that they prefer “probabilistic robustness” over “real-world” robustness, but use both at different places which could be confusing for readers.
- I do not believe that probabilistic robustness/real-world robustness is significantly different from corruption robustness as stated in the text: if we knew the distribution p_\rho corresponding to a specific type of corruptions (e.g., blur), then corruption robustness is a Monte Carlo sampling approach to estimate probabilistic robustness wrt. p_\rho. Or am I missing anything?
- The introduction is a bit fragmented as the authors try to discuss relevant related work. I would suggest moving this to a separate related work section to make the introduction more readable.
- I am wondering whether the assumption that a multivariate distribution can be transformed to be normal is a restriction. I am thinking, for example, about natural corruptions. Is it meaningful to assume that these can be transformed to a multivariate Gaussian? For, e.g., salt and pepper noise or occlusions or something like that I am not sure. I feel a discussion would be useful. More importantly, I think it would be interesting to have some concrete applications where this is the case - while Gaussian noise on MNIST is frequently considered, it is unclear to me how reasonable exponential or lognormal noise on Iris really is.
- The authors state that comparison to other notions of robustness is not meaningful. Many robustness techniques use accuracy-like metrics instead or in addition to distance metrics. Accuracies should be reasonably comparable to probabilities. So I do not agree with this argument to not compare with anything else.
- Figure 4 seems unnecessary given table 1.
- In 4.1 the authors again say that in real applications the distribution has to be fixed. I really think the paper would benefit from concrete applications and examples.
- The authors also highlight that previous work has problems with high input dimensionality (this is repeated in 4.3), but the proposed methods seems very slow even for 5x5 MNIST images. So it is unclear how the method is scalable (both in input dimension and in number of trees).
- In Figure 5 what is the y axis?
- For 4.4 I would appreciate a more realistic example. It is hard to imagine whether these distributions make sense for the Iris dataset.
- There is little ablation in terms of decision tree or random forest size.
- What happens if the decision tree is trained on the noisy images (similar to adversarial training)?

Conclusion:
I do not think that this paper is very interesting for TMLR readers. Generally, I appreciate the simplicity of the method, but the limitation of previous work in high input dimensionality is not really addressed. Moreover, the method is limited to normal or similar distributions and a discussion/argument/experiment of relevance to real problems is missing.

---

> ### Author Response · Authors · 2023-01-18
> **Response to Reviewer xKGp Part 1**
>
> We thank the reviewer xKGp for the time and the feedback, and for the opportunity to perform a revision on the submitted manuscript "Quantifying probabilistic robustness of tree-based classifiers against natural distortions". Please find below a response to the comments.
>
> - For me to recommend acceptance, besides addressing some of the writing and experiment concerns, I would really like to see a relevant application or at least an abstraction thereof. Gaussian noise on (downscaled) MNIST or IRIS is in my opinions not interesting enough.
>
>       We have added an experiment with a real-life dataset ("Using the ADAP learning algorithm to forecast the onset of diabetes mellitus") in Section 5.4 in the revised version of the manuscript to show a relevant application of our method. The dataset contains 768 data points with 8 variables and a binary outcome variable. We eliminated data points that contain unrealistic numbers which reduced the number of data points to 724. We further removed one variable since it only had a minor impact on the outcome, resulting in 7 input variables. Next, we computed the rank correlations between the variables and estimated the distributions of the input features based on histograms which showed that the variables seem to either follow a normal distribution or an exponential distribution. To estimate the uncertainty of the single data points, we prescribed a signal-noise ratio factor of 1/10 to scale the variance of the estimated best fitting distributions. In a real setting, the estimate of the uncertainty would have to come from information from the manufacturer of the devices used for the measurements. Finally we trained an XGBoosted classifier, where we achieved an accuracy of approximately 75% on the test samples (90/10 train/test-split) and computed the probabilistic robustness of test samples.
>
> - Regarding the first paragraph: in the adversarial robustness community, many papers actually report something like robust accuracy (accuracy on adversarial examples) instead or in addition to the distance to the adversarial example.
>
>       In the paper "Robustness may be at odds with accuracy" (Tsipras et al. 2018), the authors compare standard and adversarially robust accuracy of classifiers, therefore looking at the robustness from a model perspective. Our proposed method looks at individual data samples and their robustness against distortions in contrast. Comparing these two measures therefore does not appear to be meaningful.
>
> - The second paragraph about Cohen et al. feels a bit unconnected and misplaced.
>
>       We have now restructured the paper by separating the first Section into two sections (Introduction and Related Work).
>
> - In equation (2), I am wondering whether there is $p_\mu$ missing? Currently, the integral runs over $R^N$ and is not weighted by the underlying noise distribution $p_\mu$. Also, why not invert f’ and define $R_\mu$ directly as equation (2) for simplicity?
>
>       We thank the reviewer for spotting this error and apologize for the mistake. We have now revisited the math behind the method and corrected Equation 2 by weighing the integral with the probability density function in the revised version of the manuscript.
>
> - The authors state that they prefer “probabilistic robustness” over “real-world” robustness, but use both at different places which could be confusing for readers.
>
>       Where we use the term real-world-robustness we refer to the paper of (Scher & Trügler, 2022). As they use that term, we use the term to describe the definition in their paper. For our paper, however, we denote the robustness notion in this manuscript as probabilistic robustness, as we found in the previous review round that the term "real-world-robustness" can confuse readers.
>
> - I do not believe that probabilistic robustness/real-world robustness is significantly different from corruption robustness as stated in the text: if we knew the distribution $p_\rho$ corresponding to a specific type of corruptions (e.g., blur), then corruption robustness is a Monte Carlo sampling approach to estimate probabilistic robustness wrt. $p_\rho$. Or am I missing anything?
>
>       The two definitions, corruption robustness and probabilistic robustness, are indeed similar to each other and share common traits. However in Hendrycks & Dietterich (2019), corruption robustness is defined for the whole classifier ("[...] corruption robustness measures the classifier’s average-case performance on corruptions C" [...]), whereas our definition of probabilistic robustness is for single test points. Our approach is not so much concerned of finding some form of mean robustness of a classifier (which would be useful if one wants to compare classifiers), but of finding the robustness of an individual prediction, given a fixed classifier.

---

> > ### Author Response · Authors · 2023-01-18
> > **Response to Reviewer xKGp Part 2**
> >
> > - The introduction is a bit fragmented as the authors try to discuss relevant related work. I would suggest moving this to a separate related work section to make the introduction more readable.
> >
> >       We thank the reviewer for this suggestion to make the manuscript more readable. We have now restructured the paper by separating the first Section into two sections (Introduction and Related Work), to improve the reading flow.
> >
> > - I am wondering whether the assumption that a multivariate distribution can be transformed to be normal is a restriction. I am thinking, for example, about natural corruptions. Is it meaningful to assume that these can be transformed to a multivariate Gaussian? For, e.g., salt and pepper noise or occlusions or something like that I am not sure. I feel a discussion would be useful. More importantly, I think it would be interesting to have some concrete applications where this is the case - while Gaussian noise on MNIST is frequently considered, it is unclear to me how reasonable exponential or lognormal noise on Iris really is.
> >
> >       The noise on the MNIST data was chosen for illustration purposes only. We have now added a more realistic application in Section 5.4 of the revised version of the manuscript, where the uncertainty for 4 variables is best modelled by a normal distribution and for the remaining 3 variables, the uncertainty is best modelled by an exponential distribution. The uncertainty, e.g., of measured physical quantities does not necessarily have to follow Gaussian noise, especially if the quantity of interest is derived via a transformation from another quantity (such as chemical concentration, which is derived via a logarithmic law from the measured transmittance). Salt and pepper noise might indeed not be able to be transformed into a multivariate Gaussian, but it is not a reasonable uncertainty assumption outside the domain of computer vision. Especially in tasks that use tabular data measured by physical devices, the uncertainty will - at least in most cases - follow an uncertainty that can be transformed into a multivariate Gaussian. We have now added a discussion on this in the conclusion section.
> >
> > - The authors state that comparison to other notions of robustness is not meaningful. Many robustness techniques use accuracy-like metrics instead or in addition to distance metrics. Accuracies should be reasonably comparable to probabilities. So I do not agree with this argument to not compare with anything else.
> >
> >       Most accuracy-like metrics correspond to the whole classifier. In contrast, our proposed method looks at individual data samples and their robustness against distortions. Comparing these two measures therefore does not appear to be meaningful, as they deal with two different problems - measuring the robustness of a classifier over the test set, with the goal of comparing it to the robustness of other classifiers, compared to our problem statement of measuring the robustness of a single sample. The latter can also be done by measuring the distance to the closest counterfactual. This is indeed interesting, but a comparison of probabilistic robustness with the distance to the closest counterfactual has already been done by (Scher & Trügler, 2022) on two datasets, and they showed that they measure different things. We therefore, do not feel that adding such a comparison here would add any additional insights. We do now, however, mention the comparison done by Scher & Trügler at the beginning of Section 5.
> >
> > - Figure 4 seems unnecessary given table 1.
> >
> >       We agree that the Figure does not offer additional insights. We have therefore removed the Figure from the revised version of the manuscript.

---

> > > ### Author Response · Authors · 2023-01-18
> > > **Response to Reviewer xKGp Part 3**
> > >
> > > - The authors also highlight that previous work has problems with high input dimensionality (this is repeated in 4.3), but the proposed methods seems very slow even for 5x5 MNIST images. So it is unclear how the method is scalable (both in input dimension and in number of trees).
> > >
> > >       Equation 3 shows the number of created boxes, which gives an idea about the scaling of the method. A higher input feature dimension creates more boxes and also that a more detailed separation of the input space creates more boxes. While this means that the computational load increases significantly with increasing dimensionality, the results will always be exact if using the complete method, or the error will have a hard lower bound if our approach for runtime improvement is used. In the revised version of the manuscript we have, however, changed  "The approach is based on the recently introduced measure of ``real-world-robustness'', which works for all black box classifiers,  but is only an approximation and only works if the input dimension is not too high, whereas our proposed method gives an exact measure." to ``The approach is based on the recently introduced measure of ``real-world-robustness'', which works for all black box classifiers, but is only an approximation, whereas our proposed method gives an exact measure.'' in the abstract in order to avoid giving the impression that our method solves the problem with high dimensionality.
> > >
> > > - In Figure 5 what is the y axis?
> > >
> > >       We apologize for the confusion. The y-axis does not have a meaning in the plot. As we changed the example of the according Section (5.4), we also removed the Figure from the revised version of the manuscript.
> > >
> > > - What happens if the decision tree is trained on the noisy images (similar to adversarial training)?
> > >
> > >       In case a tree-based classifier is trained on different images, the method itself does not change. The input features of the test images would likely have a different uncertainty distribution or different parameters and the classifier would be different as well. This might of course affect the robustness of the test sample against noise in the images. Robust training methods are indeed an important topic, but they are, however, not the topic of our paper. We want to show how the robustness of a single test point can be measured, not what affects this robustness. Our method works far any trained tree-based classifier, independent of the exact training method used.

---

> > > > ### Comment · Reviewer_xKGp · 2023-01-19
> > > > **Another question**
> > > >
> > > > Regarding the new application, I am confused how exactly noise is estimated and applied. In the text, it is stated that the measurement noise is unknown and has to be estimated, which is reasonable. Then, this "uncertainty" (as it is referred to in the text) is estimated across examples where it turns out that most features are normally distributed.
> > > >
> > > > I feel that this confuses the distribution of features across examples with the unknown noise distribution. For example, in an example with only one input feature stemming from an unknown sensor, the value of the feature can be normally distributed across examples but the noise distribution does not have to be normal. Let's say human height is measured traditionally by standing at the wall and putting a book on one's head. Human height is roughly Gaussian, but the sensor noise distribution might not be Gaussian - it could be skewed to one side because height is hardly underestimated because the book cannot be _inside_ the head.

---

> > > > > ### Author Response · Authors · 2023-01-23
> > > > > **Reply to "Another question"**
> > > > >
> > > > > We completely agree with you that the noise distribution does not have to be the same as the distribution across samples.
> > > > > In the dataset we used, we unfortunately do not have the details of the instruments used to measure the data (and we are not aware of any comparable open dataset where this would be the case). Therefore, in our opinion, the best guess for the distribution of the noise is to assume a signal-to-noise ratio, and assume that the distribution across samples scaled by this ratio is a reasonable assumption for the noise distribution. In a practical setting, to get exact answers, in-depth domain knowledge would be required to know the uncertainty of the samples.
> > > > > The purpose of our paper is to demonstrate how probabilistic robustness - given the uncertainty distribution of the samples - can be computed in an exact way for tree-based classifiers. We believe that the current example - given the constraint that we are not aware of any open dataset that has included uncertainty information for all features - is enough to convey the main point of our paper.

---

> > ### Comment · Reviewer_xKGp · 2023-01-19
> > **Thanks for response**
> >
> > I appreciate the authors response and their updates to the paper, in particular with a more realistic example. I wanted to comment on some parts of the response:
> >
> > Adversarial/corruption robustness metrics: Both adversarial and probabilistic robustness have a point-wise notion and a population-wise. In adversarial robustness the point-wise evaluation is "can I find an adversarial example that fools the classifier in the vicinity of x" (this is usually 0 or 1, i.e., yes or no). When computing robust accuracy, we average across these point-wise results. The same applies to probabilistic robustness. In corruption robustness, this could also be applied: the point-wise estimate is the fraction of corrupted images that cause mis-classification; the population-wise is then the average of that. From that perspective, I still believe that it is not justified to say that probabilistic robustness and these other robustness metrics are inherently not comparable. Sure, they measure different aspects of robustness, but comparison is possible and surely interesting.

---

> > > ### Author Response · Authors · 2023-01-23
> > > **Reply to "Thanks for response"**
> > >
> > > We do agree with the reviewer that a comparison would be possible and interesting. We are, however, not convinced that adding such a comparison would significantly change the scientific contribution and impact of our paper, since they measure different aspects of robustness. The core point of our paper is to describe how probabilistic robustness can be computed in an exact way on tree-based classifiers.

---

### Author Response · Authors · 2023-01-18
**General Response**

We thank the 3 reviewers for their thorough review of our manuscript. We have now uploaded a revised version of the manuscript, addressing the reviewers comments with tracked changes. Please find below a short summary of the main changes. For detailed point-by-point responses to the reviewers' comments see the direct replies to the reviewers below.

1.) Section 1: We have separated the Introduction into two Sections, 1. Introduction and 2. Related Work

2.) Section 1: We extended the Introduction Section to make our contribution more clear.

3.) Section 1: We have clarified the notation in Equation (2).

4.) Section 2: We have extended the Related Work Section to emphasize the difference between our measure of robustness and the measure that is used in other papers.

5.) Section 5: We expanded the results Section with a relevant example, using the Pima-Indians-Diabetes dataset, see Section 5.4.

6.) Section 5: We have slightly expanded the Conclusion Section to make our contribution more clear.

---

> ### Author Response · Authors · 2023-01-23
> **General Response**
>
> We thank the reviewers for their remarks on our updated manuscript. Please note that we have uploaded an updated version of our manuscript with tracked changes in reponse to the new remarks by the reviewers xKGp and 6sLD.
>
> Please find below a short summary of the changes.
>
> 1.) Section 1: We have made a small addition in the second to last paragraph of Section 1 by adding "(e.g., safety critical applications or medical applications)" to emphasize where our proposed method is preferable over approximate methods.
>
> 2.) Section 3.2.1: We extended the discussion about the robustness bounds of our method

---

### Decision · Action_Editors · 2023-03-14

**Recommendation:** Reject

**Comment:**

The authors develop a method for computing robustness certificates of tree based classifiers under a noise model that is Gaussian or can be transformed to be Gaussian.

Overall, the paper's contributions do not generate interesting insights either on the theoretical, computational, modeling or algorithmic front due to the reasons discussed above. Furthermore, the claim of Gaussians being "natural distortions" is misleading.

Hence I recommend rejection

**Audience:**

The findings of the authors are unlikely to be interesting to readers of TMLR, for the following reasons:
1. As pointed out by several reviewers, the key insight(decomposing the classifier into decision reasons and integrating a Gaussian over those regions) is easily derived from first principles and does not add any insights from a computational/theoretical perspective.
2. The model of noise used in the paper is restrictive and does not cover most interesting forms of probabilistic corruptions encountered in practice. Furthermore, the advantage of an exact certificate is negated by the fact that the noise model can never actually be verified to be precisely Gaussian.

**Claims And Evidence:**

The authors develop a method to compute robustness of tree based classifiers under specific probabilistic models of corruptions of the inputs, specifically, normally distributed corruptions or those that can be transformed into normally distributed corruptions.

The algorithm proposed by the authors is sounds and the experiments support the same.

However, the claim made by the authors around "natural distortions" is misleading, since many real-world sources of noise or perturbations to input data are unlikely to be expressible as a normal distribution (for example distributions with heavy tails, or mixed discrete and continuous perturbations).